# Environmental Survey of the Distribution and Metal Contents of *Pteris vittata* in Arsenic–Lead–Mercury-Contaminated Gold Mining Areas along the Bone River in Gorontalo Province, Indonesia

**DOI:** 10.3390/ijerph19010530

**Published:** 2022-01-04

**Authors:** Nurfitri Abdul Gafur, Masayuki Sakakibara, Satoru Komatsu, Sakae Sano, Koichiro Sera

**Affiliations:** 1Regional Planning Agency—Research and Development of Bone Bolango, Bone Bolango 96562, Indonesia; 2Faculty of Collaborative Regional Innovation, Ehime University, Matsuyama 790-8577, Japan; 3Research Institute for Humanity and Nature, Kyoto 603-8047, Japan; 4School of Global Humanities and Social Sciences, Nagasaki University, Nagasaki 852-8521, Japan; skomatsu@nagasaki-u.ac.jp; 5Faculty of Education, Ehime University, Matsuyama 790-8577, Japan; sano.sakae.mm@ehime-u.ac.jp; 6Cyclotron Research Center, Iwate Medical University, Takizawa 020-01673, Japan; ksera@iwate-med.ac.jp

**Keywords:** artisanal and small-scale gold mining, *Pteris vittata*, absorption, arsenic, mercury, lead, Bone River, Gorontalo Province, Indonesia

## Abstract

In this paper, we report ecological and environmental investigations on *Pteris vittata* in the As–Pb–Hg-polluted Bone River area, Gorontalo Province, Indonesia. The density distribution of *P. vittata* decreases from around the artisanal and small-scale gold mining (ASGM) site to the lower reaches of the Bone River, and it is rarely found near Gorontalo City. The maximum concentrations of As, Hg, and Pb recorded in the soil samples were 401, 36, and 159 mg kg^−1^, respectively, with their maximum concentrations in *P. vittata* recorded as 17,700, 5.2, and 39 mg kg^−1^, respectively. Around the ASGM sites, the concentrations of As, Pb, and Hg in *P. vittata* were highest in the study area. These data suggest that *P. vittata*, a hyperaccumulator of As, may be useful as a bioindicator for assessing environmental pollution by Pb and Hg.

## 1. Introduction

Artisanal and small-scale gold mining (ASGM) is one of the largest emitters of mercury (Hg) into the environment in the world. In ASGM, miners use elemental Hg to extract gold (Au) from finely crushed rocks or Holocene sediments, forming an Hg–Au amalgam. This amalgam is then burned, releasing Hg as vapor and leaving impure Au residue. The adverse effects on the health of miners are severe, with exposure to Hg causing neurological damage and other health issues. The people residing near the ASGM sites are also affected by the Hg contamination of water, air, and soil and by the effects of Hg on ecosystems, such as fishes and plants. Approximately 15 million people in ASGM sites in over 70 countries use Hg to refine Au, causing Hg pollution [1].

In Indonesia, over 300,000 ASGM miners work at approximately 1000 informal sites. These miners are either displaced farmers or fishermen who have very few options in terms of alternative livelihoods. These rural communities depend on the ASGM for income. The practice of using Hg to extract Au was banned by the Indonesian government in 2014 [2,3]. However, ASGM mining has released hundreds of tons of Hg into the water, soil, and air. Thus, despite the passage of laws against the use of Hg in Indonesia, it has not been effective [4].

According to the data of the Ministry of Forestry and Mining of Gorontalo Province in 2012, Gorontalo Province has a large number of ASGM sites, especially in the Bone Bolango Regency, where the largest number of ASGM workers work [5]. According to Arifin et al. (2015), Gafur et al. (2018), and Lihawa and Mahmud (2019), in Gorontalo Province, the Hg concentrations in the hair of inhabitants are higher than those of the inhabitants of a non-ASGM area; a fish had Hg levels above the threshold limit by WHO/ICPS 1990 [6], and those of river water and sediment are also polluted by Hg [7,8,9].

In recent years, although many researchers have focused on Hg in environmental assessment studies at ASGM sites, little attention has been given to the other toxic metals associated with Au deposit formation. In particular, Neogene epithermal gold deposits in the Circum-Pacific Rim [10] are known to be associated with various heavy metals, such as As, Pb, Cu, and Zn. This suggests that ASGM of epithermal Au deposits is accompanied by exposure to various naturally derived harmful heavy metals in addition to anthropogenic pollution by Hg.

Generally, higher plants find it extremely difficult to survive in environments that are highly contaminated with As. However, *Pteris vittata* is a fern species with a hyperaccumulation ability for As, and it can accumulate up to approximately 28 g/kg As in its fronds [11,12]. It can also be used in the study of the bioaccumulation and biomonitoring of other heavy metals. Phytoremediation using *P. vittata* has been recently developed as an environmentally friendly and cost-effective technology to remediate heavy metal-contaminated environments [13].

*Pteris. vittata* is known to be an accumulator of metals in laboratory experiments. For example, Fayiga et al. demonstrated the capability of *P. vittata* to hyperaccumulate arsenic from soils in the presence of Cd, Ni, Pb, and Zn in a greenhouse study [14]. The novelty of the research is that it examines whether *P. vittata* acts as an “indicator plant” in severely contaminated soils in ASGM and examines the potential to remove Pb, As, and Hg.

The potential of *P. vittata* to remediate contaminated soils in China has also been empirically examined [15,16,17]. For example, Lei et al. examined the remediation efficiency of *P. vittata* in As-contaminated soils through a 2-year field experiment [15]. Since previous studies indicate the potential of *P. vittata* as an accumulator of metals, limited studies have examined the role of *P. vittata* as an accumulator of metals in the case of ASGM locations. Soils of ASGM locations are usually polluted by historical mining operations, and the potential of bioaccumulators of *P. vittata* are influenced by the level of pollutants and climatic and geographical conditions. Further empirical studies are necessary in ASGM locations to examine the potential of accumulating metals from contaminated soils. Therefore, the present study yields important policy implications for achieving sustainable livelihoods through adopting environmentally friendly technologies.

In this study, we conducted a field survey of the distribution of the population density of *P. vittata* and analyzed the metal concentrations of the samples of soils and *P. vittata* in ASGM areas polluted by multiple metals, such as Hg, As, Zn, and Pb, around the Bone River, Gorontalo Province, Indonesia. We discussed the correlations between the distributions of *P. vittata* and the metal concentrations between the soils and plants.

## 2. Materials and Methods

Over the past decade, mine tailings and wastewater from the two ASGM sites located upstream of the Bone River and smelted slag from the sites along the valley have been dumped along the river, and they are mixed into the river water.

The Bone River is in the eastern part of Gorontalo, Indonesia, within the Bone Bolango Regency (0.27°–1.01° N by 121.23°–122.44° E). The regency has an area of 1985 km^2^, and altitudes can reach up to 1500 m. The two ASGM sites in this region are Motomboto and Mohutango. *P. vittata* and soil samples were collected along the Bone River from downstream to upstream close to the ASGM areas (Figure 1).

### 2.1. Distribution of the Population Density of P. vittata and Sampling of the Plants and Soils

We mapped the distribution of the population density of *P. vittata* along the Bone River from the ASGM area to Gorontalo City. Plant population density was determined using the quadrat counts method [18] by counting the plants on the slopes along the Bone River; during this process, the locations were recorded by the global positioning system and mapped using ArcMap v. 10.3 (Esri Company, Redlands, CA, USA).

In all, 37 plant samples were collected from the field from upstream to downstream along the Bone River at appropriate intervals, and one whole plant was taken at each point. The plant tissue we used for analysis was the fronds. Soil samples (*n* = 20) were collected from depths of 10–20 cm from the *P. vittata* collection points. The soil samples were dried (120 °C for 48 h) with a ventilated oven ISUZU drying oven Model 2-2045 and stored in sterilized plastic bags prior to analysis. The dried soil samples were sieved using a 2 mm sieve for the removal of larger organic components, and the smaller components were removed using tweezers. The samples were ground to a powder in a planetary ball mill (Varian PM-2005 m, Osaka Chemical Co., Ltd., Osaka, Japan) and homogenized before analysis by particle-induced X-ray emission (PIXE) at the Cyclotron Research Center, Iwate Medical University, Japan. All the samples were imported into Japan under a permit issued by the Ministry of Agriculture, Forestry, and Fisheries, Japan, in accordance with the Plant Protection Law.

Our sampling procedures precisely estimated the correlation between contamination in the soil samples and uptake by *P. vittata* because the soil samples and *P. vittata* were simultaneously collected at the same location.

### 2.2. Chemical Analysis

For the PIXE analysis, the homogenized soil samples were mixed with Pd–C as an internal standard at a ratio of 3:1, set up onto a thin-film holder by using a collodion solution, and bombarded with a 2.9 MeV proton beam energy from a cyclotron [19]. PIXE is a valuable ion-beam analyzer that has proton beams of 1–4 MeV energy for the analysis of geological and biological samples.

The plant samples were washed with deionized water and dried in a ventilated oven at 80 °C for 48 h. The dried leaves were ground to a powder in a ball mill and prepared for analysis as follows: the sample powder (20 mg) was digested with 1 mL HNO_3_ on a hotplate at 80–100 °C for 2 h and then cooled to room temperature; subsequently, deionized water was added to obtain the final volume required for analysis. The As, Hg, and Pb concentrations in the plant samples were determined by inductively coupled plasma–mass spectrometry (ICP–MS) performed using a Varian 820-MS instrument (Agilent Technologies, Santa Clara, CA, USA) at the Integrated Center for Sciences, Ehime University, Japan. The flowchart of the methodology is illustrated in Figure 2. Due to the characteristics of PIXE, analytical errors are potentially larger if the concentration of metals is less than the minimum threshold (less than 10 ppm). However, PIXE is suitable because sample preparation is easier, and a large number of samples can be examined for a relatively short duration.

### 2.3. Quality Control and Statistical Analysis

The US National Institute of Standard and Technology (NIST) CRM 1643f [20] and multielement standard XSTC-13 from SPEX CertiPrep (Metuchen, NJ, USA) [21] were used as the calibration standards, and the certified reference material NIST SRM 2782 [22] was used as the check standard for the As, Hg, and Pb analyses of the soil samples. The analytical accuracy and precision of the plant analyses were verified using National Institute for Environmental Studies (NIES) No.9 Sargasso [23].

The significant differences among the sampling sites were determined using Stata 16.1 (StataCorp, College Station, TX, USA) and OriginLab v. 9.60 program for Windows.

## 3. Results and Discussion

The distribution of the population density of *P. vittata* in the study area (shown in Figure 1) is shown in Figure 3. The densest overgrowth of *P. vittata* is observed along the sunny slopes and valleys around the Motomboto ASGM site (Figure 4). The plant reaches a maximum height of approximately 80 cm. The density of *P. vittata* decreases from the Motomboto and Mohutango ASGM sites to Gorontalo City. It is rarely found in the western Bone Bolango Regency. The plants form communities along rivers and on slopes in slightly humid environments. The distributions, each more than 50 stocks/m^2^, are the highest around the Motomboto and Mohutango ASGM sites (Figure 4). The distribution becomes roughly lower toward the downstream side and becomes zero on the eastern side near Gorontalo City.

### 3.1. As, Hg, and Pb Concentrations in Soil and P. vittata

The soil concentrations in this study and regulatory guidelines for As, Hg, and Pb as per the US Environmental Protection Agency (EPA) (1993) [24] are listed in Table 1 and Table 2. Large differences were found in the As and Pb concentrations of the soil samples. These differences indicate that soil samples of the study area were relatively heterogeneous in terms of metal concentrations. The As and Pb concentrations were in the ranges of not detected (ND)–36,500 and 8–11,400 mg kg^−1^, respectively. As and Pb concentrations exceed the criteria set by the US EPA 1993 (Table 2). The overall tendency is that the concentration of each metal decreases from the ASGM site toward the downstream side.

The concentrations of As, Pb, and Hg in the samples of *P. vittata* and soil samples are summarized in Table 1 and Table 3, and their geographical variations are shown in Figure 5. Table 1 shows that there were 17 sampling points that were over the regulatory limit (Table 3) for the concentration of arsenic and two sampling points for the Pb concentration. This result indicates that the soils in this area were naturally polluted by As and Pb because of the weathering of the hydrothermally altered mother rocks. On the other hand, the maximum Hg concentration in the soil was 36 mg kg^−1^, which was relatively low but tended to increase toward the ASGM sites.

The concentrations of As, Pb, and Hg in *P. vittata* were in the ranges of ND–17,700, ND–39 mg kg^−1^, and ND–5.2, respectively. The ND data correspond to a situation in which the concentration is lower than the detection limit of the analytical instrument.

### 3.2. Relationship between the Distributions of the Population Density of P. vittata and the As Concentration in Soil

There is generally a positive correlation between the density distribution of *P. vittata* and the As concentration in soil. This correlation is also consistent with *P. vittata* being found in As-contaminated areas [25]. *P. vittata* grows actively in a neutral or slightly alkaline environment [11,25]. Although the pH of the soil samples was not measured in this study, it is likely that the soil pH is neutral or weakly alkaline at many sampling points.

### 3.3. Metal Pollution and Uptake by P. vittata along the Bone River

We found that *P. vittata* absorbed large amounts of As, reaching a maximum concentration of 17,700 mg kg^−1^ (Table 3). The rocks around the ASGM sites in Gorontalo are Pliocene altered andesites containing sulfide minerals, mainly pyrite. At present, most of the sulfide minerals in the altered andesitic rocks on the ground surface have been decomposed by weathering under tropical weather conditions. The As concentration in pyrite is not clearly known, but since weathered soil contains a certain amount of As, it is presumed that pyrite also contains a similar amount of As.

The results of this study show dispersion and no significant correlation between *P. vittata* and As concentration in the rhizosphere soil (Figure 6). The results of previous laboratory and field studies on *P. vittata* indicate that the removal efficiency of *P. vittata* depends on the ratio of soluble As and the pH range of the soil [12,13,25]. The results of our study suggest that the ratio of soluble As and pH of the soil in the study area may be diverse.

Generally, Hg accumulation in the soil is related to deposition caused by anthropogenic activities through the biosphere or atmosphere [26]. The ASGM activity includes techniques, such as crushing and mixing the raw materials with Hg, and the waste produced after amalgamation also contains a large amount of Hg [27,28,29,30]. We found that the Hg uptake by *P. vittata* in the mining areas was higher than that in the downstream area. The source of the high concentration of Hg in the ASGM area was mainly Au extraction activities. Hg could be transferred in forms such as methylmercury in the ecosystem [24] and was then absorbed by *P. vittata*. Based on the ability to absorb Hg, *P. vittata* is considered to have the potential for use as a bioindicator for Hg [31,32].

The maximum soil concentrations of Pb in this study area are higher than the limit prescribed by the US EPA for agricultural soil [33]. *P. vittata* had a maximum Pb concentration of 39 mg kg^−1^ that tended to be considerably higher around the ASGM sites. These results indicate the possibility of using this plant as a bioindicator for Pb in contaminated areas.

The correlations among As, Hg, and Pb concentrations in *P. vittata* are shown in Figure 6. Hg is of anthropogenic origin, whereas As and Pb are of natural origin. Despite their different origins, they generally show a positive correlation. According to the results of Spearman’s rank correlation tests, a positive and statistically significant relationship was identified among As and Hg (10% level), As and Pb (1% level), and Hg and Pb (10% level). This finding suggests that Hg is methylated over time and exposed to the ecosystem.

The correlation between As, Hg, and Pb in *P. vittata* and soil samples is shown in Figure 7. With regard to the Hg concentration, the results of Spearman’s rank correlation tests between the plant and soil samples indicate a negative and statistically significant relationship at the 5% level. This result suggests that the locations where the Hg contamination of soils is higher show lower accumulation in the plant samples. The results are uncertain because of the limited number of samples, and further investigation is needed to determine if Hg absorption saturation has been attained at the study sites.

The findings raise concern for biodiversity in the local environment. Since *P. vittata* can adopt and survive in metal-contaminated soils, there is a concern that it could be an invasive species, which thereby influences local biodiversity. Rathinasabapathi discusses the competitive ability and invasiveness of several ferns [34]. As there is no empirical evidence about the invasive nature of *P. vittata* in the surveyed areas, further studies are required to understand the influences of expanding *P. vittata* for other local species.

The study did not examine the mobile/bioavailable fraction of metals that could potentially correlate with uptake by *P. vittata*. Examining the composition of mobile fractions extracted from environmental solid samples may provide a clearer explanation of the uptake. There are two reasons why the study did not examine the composition of mobile/bioavailable fractions. First, the current study provides a relationship between uptake and total metal contamination, even though mobile/bioavailable composition is not fully investigated. More total metal contamination is identified, and more plant uptake is empirically identified. Second, even mobile/bioavailable and solid fractions are mixed in the field, providing evidence for total metal contamination and plant uptake yields important implications for policy-makers, especially in developing countries, regarding whether contamination is prevalent. At this point, the research is considered a preliminary investigation, and further research is necessary. Examining the composition of mobile fractions from environmental solid samples to show clear characteristics between metal compositions and plant uptake is beneficial to elaborate on the findings of the research.

## 4. Conclusions

We conducted ecological and environmental investigations on *P. vittata* in the As–Pb–Hg-polluted Bone River area, Gorontalo Province, Indonesia. The density distribution of *P. vittata* decreases from around the ASGM sites to the lower reaches of the Bone River, and the plant is rarely found near Gorontalo City. The maximum concentrations of As, Pb, and Hg in *P. vittata* were 17,700, 5.2, and 39 mg kg^−1^, respectively. Around the ASGM sites, the metal concentrations in *P. vittata* were the highest in the study area. These data suggest that *P. vittata*, which is a hyperaccumulator of As, may also be useful as a bioindicator for assessing the environmental pollution caused by Pb and Hg.

Since this research was limited to case studies in the ASGM regions of Gorontalo Province, Indonesia, further research is pivotal for generalizing the results. For example, a high level of mercury contamination has been reported in ASGM communities in various provinces, such as in West Java [35], West Nusa Tenggara [36], South Sulawesi [37], and Buru Island, Mollucas [38]. Further studies are necessary to determine whether *P. vittata* acts as an indicator plant in other areas. Identifying the potential and limitations of *P. vittata* as an indicator plant by extensive investigations will be helpful for policy-makers to assess the environmental pollution of respective ASGM areas.

## Figures and Tables

**Figure 1 ijerph-19-00530-f001:**
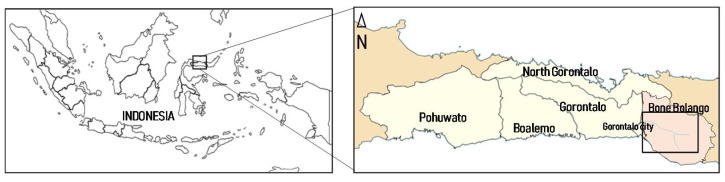
Location of Gorontalo in Indonesia (**left**) and map of Gorontalo (**right**) with the study area along the Bone River enclosed in a black rectangle.

**Figure 2 ijerph-19-00530-f002:**
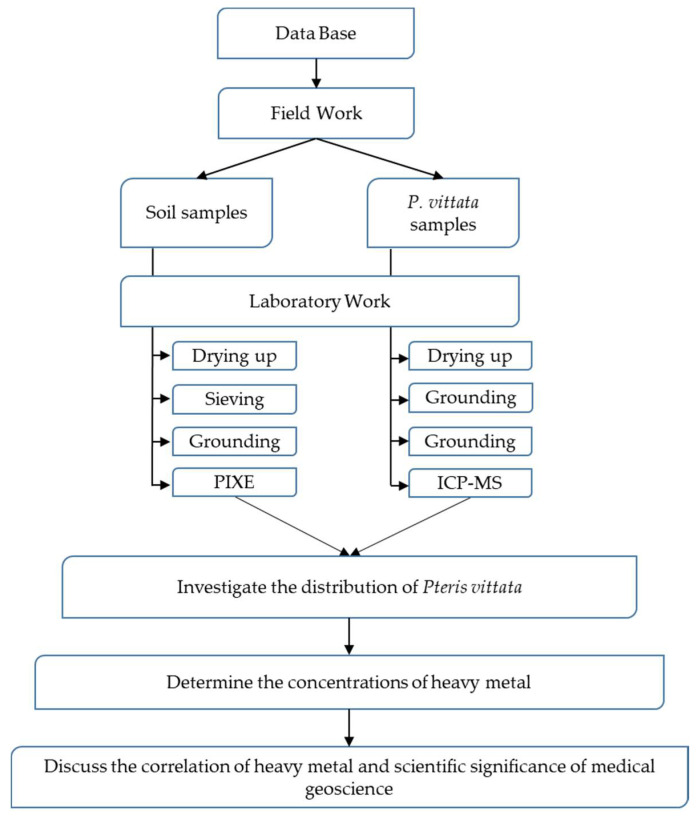
Flowchart of methodology.

**Figure 3 ijerph-19-00530-f003:**
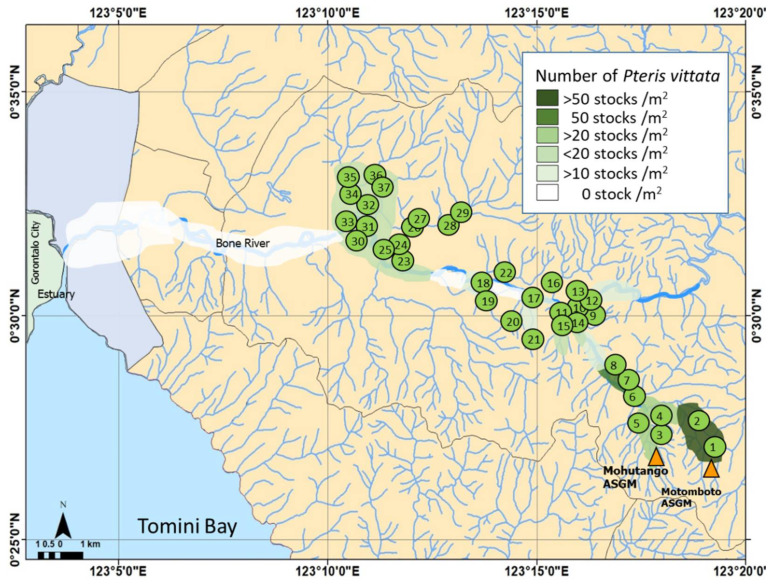
*P. vittata* distribution map and sampling point.

**Figure 4 ijerph-19-00530-f004:**
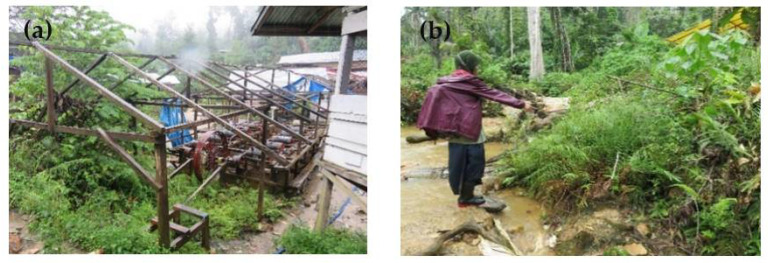
Mode of occurrence of *Pteris vittaae* around the Motomboto ASGM sites in Bone Bolango Regency, Gorontalo Province. (**a**) *P. vittata* growing in mine trommels; (**b**) *P. vittata* growing at a site contaminated with As–Pb at the Motomboto ASGM site.

**Figure 5 ijerph-19-00530-f005:**
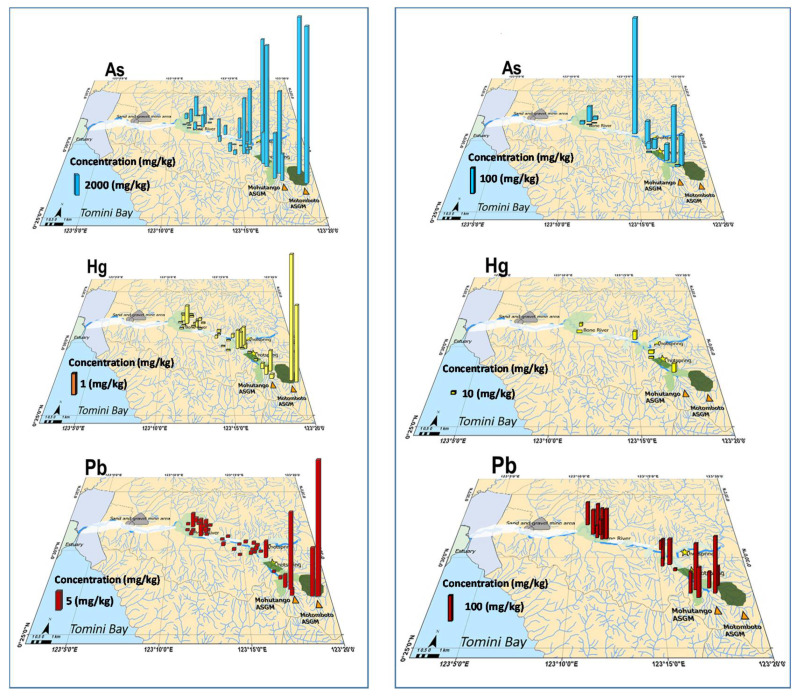
As, Hg, and Pb concentrations in *P. vittata* (**left**) and soil (**right**) samples were taken along the Bone River (mg kg^−1^).

**Figure 6 ijerph-19-00530-f006:**
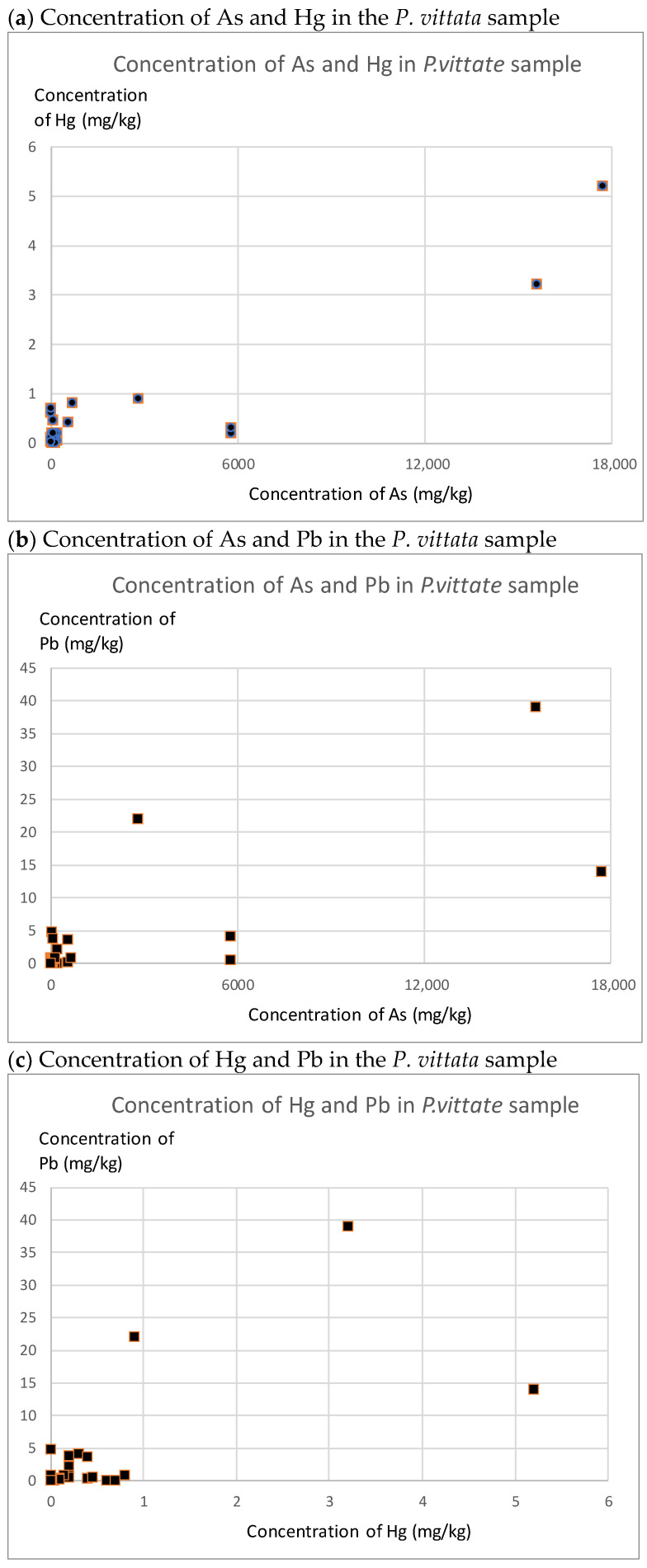
Scatter plots depicting the relationships between As, Hg, and Pb concentrations in *P. vittata* samples.

**Figure 7 ijerph-19-00530-f007:**
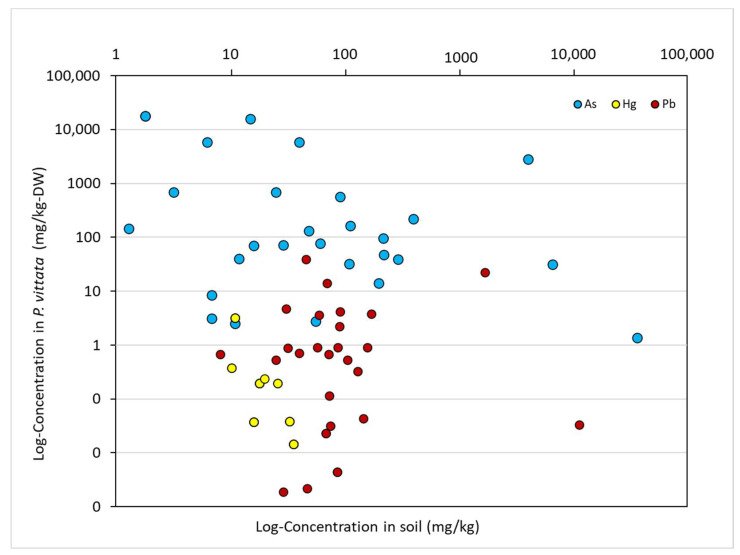
As, Hg, and Pb uptake intensity from the soil to shoot of *P. vittata*; DW: dry weight.

**Table 1 ijerph-19-00530-t001:** Concentrations of As, Hg, and Pb in the soil samples along the Bone River.

No.	Elements					
	Concentration (μg/g)		Concentration (μg/g)		Concentration (μg/g)	
	As	Error	Hg	Error	Pb	Error
1	15.0	24.0	11.0	17.0	46.0	18.0
2	1.80	4.20	ND	ND	70.0	16.0
3	ND	ND	ND	ND	90.0	31.0
4	4060	149	ND	ND	1700	178
5	92.0	8.80	10.3	17.0	60.0	32.0
6	40.0	29.0	ND	ND	8.20	18.0
7	401	21.0	33.0	19.0	29.0	59.0
8	6.30	5.50	ND	ND	92.0	21.0
9	199	14.0	ND	ND	47.0	48.0
10	110	11.0	18.0	22.0	58.0	39.0
11	11.0	7.50	ND	ND	74.0	29.0
12	3.20	11.0	ND	ND	159	42.0
14	61.0	49.0	ND	ND	25.0	33.0
15	6600	298	ND	ND	75.0	340
16	25.0	7.42	ND	ND	87.0	29.0
17	112	14.0	26.0	22.0	106	51.0
18	56.0	14.0	ND	ND	69.0	51.0
19	1.30	4.20	16.0	15.0	63.0	16.0
20	295	13.0	ND	ND	32.0	36.0
21	16.0	4.00	ND	ND	40.0	15.0
22	12.0	11.0	ND	ND	146	44.0
23	29.0	35.0	ND	ND	130	26.0
24	6.90	6.60	36.0	21.0	86.0	25.0
25	6.90	8.20	ND	ND	63.0	32.0
26	221	19.0	ND	ND	31.0	64.0
27	36,500	1376	ND	ND	11,400	735
28	49.0	29.0	ND	ND	73.0	19.0
29	218	19.0	20.0	27.0	170	65.0

ND, not detected; error, analytical error. Investigations were performed three times at each location.

**Table 2 ijerph-19-00530-t002:** As, Hg, and Pb concentrations in the soil samples taken along the Bone River in Gorontalo Province and the regulatory guidelines for As, Hg, and Pb according to the US EPA (1993).

Metal/Metalloid	Soil Concentration Range mg kg^−1^	Soil Regulatory Limits * mg kg^−1^
As	0–36,500	20
Hg	0–36	270
Pb	8–11,400	600

* US EPA 1993.

**Table 3 ijerph-19-00530-t003:** Concentrations of As, Hg, and Pb in the *P. vittata* samples taken from the Bone River area.

Sample	As		Hg		Pb	
Number	Concentrations mg kg^−1^	RSD	Concentrations mg kg^−1^	RSD	Concentrations mg kg^−1^	RSD
1	15,600	1.30	3.20	3.00	39.0	4.20
2	17,700	1.20	5.20	7.70	14.0	3.30
3	220	1.10	0.20	20.0	2.20	3.80
4	2810	5.50	0.90	5.60	22.0	4.80
5	570	1.60	0.40	4.10	0.30	1.20
6	5800	2.60	0.20	5.10	0.60	7.80
7	220	1.70	0.04	12.0	0.002	3.10
8	5800	0.50	0.30	21.0	4.10	2.20
9	14.0	0.90	0.01	10.0	0.002	1.40
10	32.0	18.0	0.20	11.0	0.90	1.50
11	2.50	2.10	0.60	5.00	0.10	3.00
12	570	1.60	0.40	4.00	3.60	1.20
13	77.0	0.40	0.45	7.00	0.50	1.60
14	32.0	0.70	0.02	24.0	0.03	5.00
15	690	1.50	0.80	3.00	0.90	1.10
16	160	0.90	0.20	4.00	0.53	1.50
17	2.80	1.10	0.01	17.0	0.02	1.20
18	140	1.30	0.04	7.10	ND	ND
19	39.0	0.90	0.14	19.0	0.90	1.70
20	69.0	0.80	ND	ND	0.70	2.50
21	39.0	2.00	ND	ND	0.04	1.10
22	71.0	3.10	0.10	8.20	0.30	2.40
23	3.10	0.70	0.01	7.20	0.004	3.10
24	8.30	2.00	0.01	6.50	ND	ND
25	47.0	1.90	0.00	0.79	4.80	26.0
26	1.30	1.33	0.02	14.2	0.03	1.30
27	130	0.90	ND	ND	0.70	0.90
28	95.0	3.10	0.20	5.89	3.80	1.80
29	8.80	0.80	0.10	6.87	0.20	8.00
30	5.40	0.90	0.01	15.5	ND	2.20
31	7.00	0.70	0.01	18.3	0.01	3.90
32	8.20	4.00	0.70	26.3	0.10	6.10
33	6.40	1.30	0.02	5.60	ND	2.40
34	95.0	0.90	0.20	3.90	3.82	2.00
35	34.0	1.20	ND	ND	0.10	17.0
36	153	0.80	ND	ND	0.90	1.40
37	2.10	1.20	0.01	15.0	ND	6.40

ND, not detected; RSD, relative standard deviation. Investigations were performed three times at each location.

## Data Availability

The data were generated during the study.

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
