# Peer review of "Environmental Survey of the Distribution and Metal Contents of Pteris vittata in Arsenic–Lead–Mercury-Contaminated Gold Mining Areas along the Bone River in Gorontalo Province, Indonesia"

_ijerph, 2022, doi:10.3390/ijerph19010530_

Round 1

Reviewer 1 Report

The following are my suggestions to improve the work - 

  1. Relevant literature in this field must be cited, specifically recent articles 2019 -2021. In the introduction, the need of the research must be highlighted.
  2. The study methodology should be described in detail, if possible through a running flowchart. 
  3. Results and Discussion section must be strengthened with better representation of the result. No need to split in two sections. 
  4. Figure 5 quality is not at par. Must be improved. 
  5. The impact of the findings on environment must be highlighted.

Reviewer 2 Report

The paper fits well to the thematic scope of the journal. The results of this study are original and may possibly be of interest for the readers working with the issues of environmental contamination.
The Manuscript could represent an interesting contribution to the scientific knowledge on the distribution of contamination and its transfer in the soil / plant system. 
Nevertheless, I have the following comments that should be taken in consideration for manuscript revision:

Anyway, Major revision is recommended before its acceptance to publication. Revisions are required to give more clarity in the methodology adopted and, in the result, discussion sections. 
The following comments should be taken into consideration for manuscript revision:
-    The manuscript, requires a general improvement in all parts including data presentation (figures) and interpretation should be much more detailed showing also advantages and disadvantages of the adopted sampling and analytical procedures 

-    The data should be integrated with metals mobile/bioavailable fractions soil need much more studies and should be stated in the manuscript. Also data obtained needs to be much more discuss.  

Comments and suggestions related specific sections of the manuscript are listed below.

The paper also needs a revision for English language, but as I am not native English I could not give specific suggestions.

Title: Improve it including Hg element as element reported on the study. Erase “heavy”: Arsenic and mercury are not heavy metals
Abstract: it well resume data of the manuscript

Introduction: it is clear in its objectives but would need some integration and general improvement reporting reference in which pteris vittate have been suggested as “indicator” plant.

Materials and Methods: it needs to be integrated with details.
a) Have been sampled a control site, uncontaminated soil, with plant samples? Such choice permits an evaluation of plant uptake ability. Sometimes hyperaccumulator are not considered indicator plant due to its extraordinary ability in metal accumulation even if the metal is detected at low  content in soil. 
b) Insert details on: soil sampling procedure; soil treatment before the analytical procedures (drying sieving?); analytical procedure used for soil characterization.
c) Add chemical physical soil characteristics (clay, sand, lime %, pH, organic matter, CSC). Such data give important details on soil propertises linked to 
d) Insert details on the metal chemical fractionations of sampled soil. Also simplified sequential extraction could give relevant information on metals chemical forms in soil and their dangerous to environmental process.
e)Details on plants sampling, numbers of plants for each sampling point, plant tissues collected (fronds, roots).
f) Check procedure for metals analysis, high temperature for sampling drying can induced the lost of volatile elements (As, Hg). In fact, most of the official procedures recommend the air drying or at low temperature (40°C).
g) The data seems not sufficient to permit such correlation and sometimes what plants can uptake is much more correlated with mobile /bioavailable fraction than on the total metal concentration in the soil. Many studies have been conducted in mining areas showing large variation of the content also due to natural geological level. 
h) Moreover figures of the scattered plots do not allow to indicate Pteris vittata as an indicator plant. Please check data and integrated with much more consideration.
Some factors can be calculated such as Bioconcentration (soil/plant ratio) and translocation factor (root/ shoot ratio) to better discuss the metal (loids) transfer from soil to plants.
Plant sampling: describe how many samples at each sampling point have been collected and what section of the plant has been collected and analyzed. 
How many replicates have been referred the error indicated in table 1?
Pleas indicate in the map the sampling points and how many over passing regulatory limits? As the results indicated a range that goes also below the limit
Results: it needs in general to be integrated with details and data as suggested in the previous section.
Discussion: it needs to be integrated with details as suggested in the previous section. Moreover other citation of paper with analog aim.

For so, I would represent the manuscript as a preliminary study investigating the metal transfer for soil to plants.

Figures and tables
Legends should be more clear.
Data indication should be more clear in replicates done and related error.
Conclusion
It needs further comments, including future investigation in the plan.
The authors do not present a truly "Conclusions" section. Authors can also suggest possible implications and applications of this knowledge and "pathways" for future work.

The described comments allow me to determine that this manuscript is not suitable to be published in the present form. 
I hope suggestions could be of help for editors and authors. 

Reviewer 3 Report

The report by Nurfitri Abdul Gafur and co-authors on the levels of trace metals in soils and Pteris vittata plant in Indonesia has been evaluated. The manuscript presents research with somewhat average scientific novelty. However, the findings may be useful for further management of local environmental issues. 

Comments:

  1. Language should be checked again.
  2. line 38-39. citation required.
  3. line 71. Why write about a river when the research was conducted on soil samples?
  4. What was the nature of the soil samples? Marshy or clay?
  5. line 80-82. Make this sentence the beginning of the materials and methods section.
  6. line 84-87. This segment looks like part of the results.
  7. what is the justification for sampling depth 10-20cm?
  8. line 159. How was the analytical error determined?
  9. line 161. check the spelling of river.
  10. What was the statistical correlation between the levels of trace metals in soil and the levels in the plant monitored. This will help to further validate the bioaccumulation claims in the conclusion section. 

Reviewer 4 Report

The paper is generally well written and organized. Their objectives are related to health concerns of potentially toxic elements. Furthermore, I made additional suggestions as detailed below:

  1. Explain in more detail for Section 2.2. Chemical analysis of the preparation and determination of soil samples for analysis of As, Hg, and Pb concentrations in soil.
  2. In Table 2. As, Hg, and Pb concentrations in the soil samples taken along the Bone River in Gorontalo Province presented the soil regulatory limitation of EPA for As, Hg, and Pb. How about the standard limit of the studied country?
  3. Modify Figure 5. Scatter plots depicting the relationships between As, Hg, and Pb concentrations in P. vittate samples to make it more clear and easier for understanding.

Round 2

Reviewer 3 Report

The authors seem to have addressed my concerns.